# Metabolic Dynamics and Sensory Impacts of Aging on Peony Mead: Insights into Nonenzymatic Reactions

**DOI:** 10.3390/foods14061021

**Published:** 2025-03-17

**Authors:** Yuqian Ban, Yanli Zhang, Yongrui Ti, Ruiwen Lu, Jiaoling Wang, Zihan Song

**Affiliations:** 1College of Horticulture, Shanxi Agricultural University, No. 1 Mingxian South Road, Jinzhong 030801, China; yuqianban@outlook.com; 2State Key Laboratory of Vegetable Biobreeding, Institute of Vegetables and Flowers, Chinese Academy of Agricultural Sciences, Beijing 100081, China; 13934604283@163.com (Y.Z.); 12316047@zju.edu.cn (Y.T.); luruiwen00june@163.com (R.L.); 3Nanjing Institute of Agricultural Mechanization, Ministry of Agriculture and Rural Affairs, Nanjing 210014, China

**Keywords:** peony mead, aging, nonenzymatic, GC-IMS, UHPLC–MS/MS

## Abstract

Peony mead, an emerging fermented beverage, has attracted attention because of its unique flavor and health benefits. The dynamic changes in sensory quality and the molecular mechanisms involved during post-fermentation are still unclear, limiting its industrial production. In this study, GC-IMS (gas chromatography-ion mobility spectrometry) and UHPLC-MS/MS (ultrahigh-performance liquid chromatography–tandem mass spectrometry) were employed to systematically analyze the variations in aroma and quality of peony mead across aging stages. During the aging process, titratable acid content increased significantly, while soluble solids and reducing sugars decreased. Total phenol content initially rose but subsequently declined. Sensory analysis demonstrated that the sweet–acid balance and polyphenol content were critical in shaping the sensory characteristics of the product. Seventeen key volatile metabolites were identified via GC-IMS, with the 2-methyl-1-propanol dimer/polymer and 3-methyl-1-butanol dimer/polymer serving as potential characteristic markers. These key volatile metabolites underwent physicochemical reactions, yielding complex and coordinated aroma characteristics. UHPLC–MS/MS analysis revealed that nonvolatile metabolites changed significantly, which were driven by nonenzymatic reactions such as redox reactions, hydrolysis, and condensation. In addition, correlation analysis identified mechanisms by which key metabolites potentially contributed to sensory properties such as floral aroma, fruit fragrance, sweetness, sourness, etc. This study provided insights into quality changes during aging and supported the development of high-quality fermented beverages.

## 1. Introduction

With the growing consumer demand for healthy drinks, fermented alcoholic products have received widespread attention due to their unique flavor, rich nutritional value, and potential health benefits [1,2,3]. According to reports, the global fermented beverage market is expected to expand steadily at a compound annual growth rate of 4.6% between 2024 and 2030. As a key segment, mead demonstrates strong growth potential, with the market projected to grow from USD 591.5 million in 2024 to USD 1395.7 million by 2032, achieving a CAGR of 11.33% during the forecast period [4,5].

Mead is a traditional fermented beverage primarily derived from honey. As one of the earliest known alcoholic drinks in human history, its origins can be traced back to the Neolithic Age. Archaeological evidence from the Jiahu site in Henan Province, China (circa 7000 BCE, approximately 8600 years ago), revealed through chemical analysis of pottery residues the presence of tartaric acid, beeswax biomarkers (such as n-alkanes and fatty acids), and fruit-specific compounds. This confirms the production of a mixed fermented drink containing rice, honey, and hawthorn fruits at that time. This archaeological discovery is corroborated by historical records. Notably, the Ming Dynasty pharmacologist Li Shizhen documented in his *Compendium of Materia Medica* (1596) the therapeutic effects of honey mead on skin diseases and respiratory ailments, revealing its dual role in ancient dietary practices and traditional medicine [6,7].

Peony (*Paeonia ostii* T. Hong et J.X. Zhang), a traditional ornamental flower in China, has also achieved significant progress in its medicinal and edible applications. Based on their distinct nutritional profiles, peony seeds are pressed into high-nutrition edible oils, while the stamens are processed into specialty teas due to their unique flavor. Notably, peony petals, rich in polyphenols, amino acids, and vitamins, are currently utilized mainly as petal tea. However, their potential for deeper development in areas (such as functional foods) remains substantial [8]. In the early stages, peony mead was fabricated using honey as the carbohydrate source and combining peony petals, demonstrating the characteristics of rich aroma, smooth taste, distinct flavor levels, and rich in a variety of bioactive ingredients (such as polyphenols, flavonoids, and organic acids), which provide unique sensory properties and potential health benefits [9].

Traditional wines and spirits rely on long-term aging to develop complex flavor profiles, while light beverages achieve fresh tastes through short-term aging [10,11]. The latter, exemplified by beer, undergoes various controlled post-fermentation aging reactions over a period of 3 to 6 weeks. Oxidation reactions catalyze the formation of fruity and caramel-like flavor compounds (with micro-oxygenation control); the Maillard reaction generates unique aromas and flavors, such as eugenol and vanillin; and limited yeast autolysis releases amino acids and other umami components, enhancing the overall complexity and depth of flavor [12]. However, existing studies on peony mead have focused mainly on fermentation process optimization or basic component analysis, lacking systematic studies on its quality change mechanism after a short-term aging process, especially the dynamic changes in metabolites and flavor complexity driven by nonenzymatic reactions [13]. This research gap limits the in-depth understanding of the quality evolution path and flavor balance mechanism of peony mead, thus hindering the optimization of industrial production processes and the development of high-value-added products.

Moreover, preliminary tests revealed that the body level is not rich enough with a weak taste after the fermentation of peony mead. After 30 days of aging under constant temperature, constant humidity, dark light, and slight oxygen, the color, aroma, and taste of the wine gradually reach the optimal level. Herein, the physical/chemical indices, sensory properties, and metabolite composition of peony mead were systematically monitored at different storage stages (M1 to M3) through a 30-day simulated aging test combined with gas chromatography-ion mobility spectrometry (GC-IMS) and ultrahigh-performance liquid chromatography–tandem mass spectrometry (UHPLC-MS/MS) technology. The quality evolution path of peony mead during the aging process was analyzed from the perspective of volatile and nonvolatile metabolites, and the contribution of dynamic changes in metabolites driven by nonenzymatic reactions to flavor complexity was revealed.

## 2. Materials and Methods

### 2.1. Materials

The *Saccharomyces cerevisiae* strain RV 171 was procured from Angel Yeast Co., Ltd. (Yichang, China). Red Phoenix peony flowers were purchased from Anhui Jinshang Flower Tea Co., Ltd. (Bozhou, China). Food-grade potassium metabisulfite was purchased from the Yiqi Department Store (Changsha, China). NaOH standard solution (0.1 mol/L), folinol reagent (≥98%), gallic acid (≥98%), anthranone (≥98%), concentrated sulfuric acid (≥98%), and standard glucose solution (≥98%) were obtained from Solebol Co., Ltd. (Beijing, China). Ammonium acetate (≥99%) was provided by Sigma Aldrich (Darmstadt, Germany), acetonitrile (≥99%) was offered by Merck (Darmstadt, Germany), and ammonium hydroxide (≥99%) and methanol were purchased from Fisher (Waltham, MA, USA).

### 2.2. Instruments

The microporous membrane filter was obtained from the source factory of Dibur brewing equipment (Yantai, China). A PHS-3E pH meter was offered by Yi Electrical Scientific Instrument Co., Ltd. (Shanghai, China). A hand-held Abbe refractometer (WZS 32C) was purchased from the Yingzhi Equipment Store (Zhengzhou, China). A high-precision liquid hydrometer was provided by Huangyu Scientific Instruments Co., Ltd. (Chengdu, China). An MJ-70-I constant temperature and humidity incubator was purchased from Gipsy Technology Co., Ltd. (Beijing, China).

### 2.3. Fermentation of Peony Mead

The peony flowers and pollen were boiled, cooled, and filtered to obtain the peony filtrate. The filtrate was mixed with various nectar to prepare peony mead. The soluble solid content was within 30 °Bx, and the pH was within 5.0. Yeast (*Saccharomyces cerevisiae*, 0.8 g/L) was inoculated, and yeast extract (0.6 g/L) was added 24 h later for nutritional supplementation. Fermentation was performed at 25 ± 0.1 °C. After being filtered through a microporous membrane filter, the honey mead was packed into a 304 stainless steel-sealed storage tank, stabilized with potassium metabisulfite (0.15 g/L), aged in a constant-temperature (15 ± 0.1 °C) and humidity-controlled (60%) incubator for 30 days after sterilization, and then sampled on days 0, 15, and 30.

### 2.4. Physicochemical Analysis

pH was measured with a PHS-3E pH meter; specific gravity was determined using a high-precision liquid hydrometer; total soluble solids were measured using a hand-held Abbe refractometer (WZS 32C); and titratable acidity was analyzed according to the GB10220-88 [14]. The reducing sugar content was quantified using anthrone colorimetry [15], while the total phenolic content was determined through the Folin–Ciocalteu method [16]. Chromatic characteristics were subsequently analyzed at 425/625 nm wavelengths employing a DigiEye Digital Imaging System. The chromatic characteristics were determined by the three parameters, *L**, *a**, and *b**, where *L** represents lightness, *a** represents red (+) and green (−), and *b** represents yellow (+) and blue (−). The values Δ*L*, Δ*a*, and Δ*b* indicate the differences in *L**, *a**, and *b** between two samples. The overall color difference (Δ*E*) is calculated based on these differences as follows:
(1)
$$\Delta E=\sqrt{{\left(\Delta L\right)}^{2}+{\left(\Delta a\right)}^{2}+{\left(\Delta b\right)}^{2}}$$


When Δ*E* < 1, the color difference is almost imperceptible; when 1 < Δ*E* < 2, the color difference can only be detected by experienced observers. When 2 < Δ*E* < 3.5, the color difference can be detected by less experienced observers; 3.5 < Δ*E* < 5 indicates a significant color difference; and when Δ*E* > 5, it means that for the observer, these are two distinctly different colors [17].

### 2.5. Colony Count

Microbial analysis was performed according to GB4789.2–2016 guidelines via the plating of serial dilutions on agar media cultured at 37 °C for 48 h [18].

### 2.6. Sensory Evaluation

Sensory attribute appearance, aroma, and taste were evaluated by a trained panel following GB/T 10220-2012 standards in a controlled environment [19,20,21]. Sensory description terms and definitions of peony mead are provided in Table A1. Grading criteria for sensory assessment are listed in Table A2.

### 2.7. GC-IMS Analysis

The volatile organic compounds (VOCs) in the peony mead were detected via a GC-IMS system (FlavorSpec^®^, G. A. S., Dortmund, Germany) and a WAX column (30 m × 0.53 mm × 1 μm, Restek, PA, USA). The peony samples (1 mL) were placed in a 20 mL headspace bottle and incubated at 60 °C for 15 min. The analysis parameters of GC-IMS were set to a total analysis time of 30 min, a column temperature of 60 °C, a carrier gas of N2, an IMS temperature of 45 °C, an injection volume of 100 µL, and a syringe temperature of 85 °C. In this study, the peak volume normalization method was utilized for the semi-quantitative analysis of volatile organic compounds (VOCs). Specifically, the relative abundance of each component was determined by calculating the ratio of its individual peak volume to the total peak volume. The formula used for this calculation is as follows:
(2)
$$Relative\;content=\frac{Peak\;volume\;of\;a\;single\;peak}{Total\;peak\;volume}\times 100\%$$


Experimental data were collected using VOCal software (G.A.S.), recording key parameters including retention time (RT), drift time (DT), peak volume (PV), and peak height (PH). The Reporter plugin was used to visualize spectral differences between samples through 3D terrain maps, 2D top views, and difference spectra, enabling the rapid identification of significantly altered peaks. The Gallery Plot plugin generated fingerprint heatmaps based on normalized peak height. For multivariate analysis, the Dynamic PCA plugin performed dimensionality reduction and clustering via principal component analysis (PCA), with the number of principal components determined by a cumulative variance contribution rate exceeding 70%. Unknown samples were classified by projecting their PCA coordinates onto clusters pre-established from reference datasets. All analyses were conducted using the built-in algorithms in VOCal software (0.4.01, G.A.S.) [22,23].

### 2.8. Non-Targeted Metabolomics and Metabolite Analysis

#### 2.8.1. Sample Preparation

Appropriate samples were slowly thawed at 4 °C and extracted via a precooled methanol/acetonitrile/aqueous solution mixture (2:2:1, *V*/*V*). After vortexing, the mixture was subjected to low-temperature ultrasonication for 30 min, allowed to stand at −20 °C for 10 min, and then centrifuged at 14,000 r at 4 °C for 20 min. The supernatant was vacuum dried and reconstituted with 100 μL of an acetonitrile–water solution (1:1, *v*/*v*) for MS analysis. After centrifugation at 14,000 r at 4 °C for 15 min, the supernatant was analyzed via mass spectrometry.

#### 2.8.2. UHPLC–MS/MS Analysis

Analysis was performed via UHPLC (1290 Infinity LC, Agilent Technologies, Santa Clara, CA, USA) coupled to a quadrupole time of flight (AB Sciex TripleTOF 6600, Shanghai, China). For HILIC separation, samples were analyzed via a 2.1 mm × 100 mm ACQUITY UPLC BEH Amide 1.7 µm column (Waters, Ireland). In both the positive and negative ESI modes, the mobile phase contained A (25 mM ammonium acetate and 25 mM ammonium hydroxide in water) and B (acetonitrile). The chromatographic method was as follows: (a) an isocratic hold at 95% B for the first 0.5 min; (b) a linear gradient decreasing from 95% to 65% B over 6.5 min (from 0.5 to 7.0 min); (c) a secondary gradient reducing to 40% B within 1.0 min (from 7.0 to 8.0 min); (d) isocratic elution at 40% B for 1.0 min (from 8.0 to 9.0 min); (e) rapid equilibration back to 95% B in 0.1 min (from 9.0 to 9.1 min); followed by (f) a 3.0-min column re-equilibration at 95% B (from 9.1 to 12.1 min). The ESI source conditions were set as follows: Ion Source Gas1, 60; Ion Source Gas2, 60; curtain gas, 30; source temperature, 600 °C; and IonSpray Voltage Floating, ±5500 V. In only MS acquisition, the instrument was set to acquire over the m/z range of 60–1000 Da, and the accumulation time for the TOF MS scan was set at 0.20 s/spectra. For auto MS/MS acquisition, the instrument was set to obtain over the m/z range of 25–1000 Da, and the accumulation time for the product ion scan was set at 0.05 s/spectra. The product ion scan was acquired via information-dependent acquisition (IDA) with a high-sensitivity mode. The parameters were set as a collision energy (CE) at 35 V with ± 15 eV, a declustering potential (DP) of 60 V (+) and −60 V (−), and isotopes within 4 Da, and there were 10 candidate ions to monitor per cycle.

#### 2.8.3. Preprocessing and Multivariate Analysis of Metabolomics Data

To convert the raw data files into mzXML format, we employed ProteoWizard software (Sourceforge, Palo Alto, CA, USA). The converted data were subsequently processed using an in-house developed program based on the R package XCMS (version 3.2) (http://www.r-project.org/, accessed on 13 February 2024) for peak detection, extraction, alignment, and integration. For metabolite annotation, OSI-SMMS (version 1.0, Guangzhou Genedenovo Biotechnology Co., Ltd., Guangzhou, China) was utilized, referencing an internal MS2 database.

For statistical analysis, principal component analysis (PCA), partial least squares discriminant analysis (PLS-DA), and orthogonal partial least squares discriminant analysis (OPLS-DA) were conducted using the R package models to identify significant metabolites between the control and treatment groups. Student’s *t*-test (threshold *p*-value < 0.05) and the variable importance in projection (VIP) score from the OPLS-DA model were combined to evaluate the significance of the metabolite data. Metabolites with a *t*-test *p*-value less than 0.05 and a VIP score of at least 1 were classified as differentially expressed. Heatmaps were created on the OmicsMart platform (Genedenovo Biotechnology Co., Ltd., Guangzhou, China) [24,25].

### 2.9. Statistical Analysis

The statistics were analyzed via one-way ANOVA using SPSS 22.0 (*p* < 0.05) (IBM, Armonk, NY, USA), with visualizations plotted in Origin X9 (Origin Lab, New York, NY, USA), OmicsMart (Genedenovo Biotechnology Co., Ltd., Guangzhou, China), and Adobe Illustrator (Adobe Inc., San Jose, CA, USA).

## 3. Results

### 3.1. Quality Characteristics of Peony Mead During Storage

#### 3.1.1. Physical and Chemical Properties of Peony Mead During Storage

During the aging process of peony mead, the changes in the physicochemical indicators (pH, SG (specific gravity), TSS (total soluble solids), TA (titratable acidity), RS (reducing sugar), TPC (total phenolic content), and color values (*a**, *b**, *L**)) were systematically investigated at three time points (M1, M2, and M3) within a 0–30-day storage period. As shown in Figure 1A, the pH and SG remained relatively stable without significant changes from M1 to M3. In contrast, TA increased significantly with increasing storage time, whereas TSS and RS gradually decreased. Moreover, the TPC initially increased but then decreased (Figure 1B). In terms of color, the *a** value showed a progressive decline, the *b** value displayed a modest rise, and the *L** value did not show significant variation (*p* > 0.05) (Figure 1C). Furthermore, Δ*E* values remained constrained within 1.0 ± 0.2 throughout the M1–M3 period (see Table 1 for statistical breakdown), as evidenced by the requirement for ISO-certified sensory panels to identify subtle chromatic deviations, confirming the fermentation matrix maintained chromatic integrity with statistically insignificant hue fluctuations, as illustrated in Figure 1D [26,27]. Microbial analysis revealed no visible microbial colonies through plate culture methods at any of the three time points (CFU < 1/mL) (Figure 1E). These findings collectively demonstrate that metabolite dynamics during aging predominantly originate from nonenzymatic chemical pathways (e.g., oxidative polymerization), with microbial metabolic contributions being statistically negligible. Specifically, TSS, RS, TA, and TPC emerged as the most storage-sensitive indicators in short-term aging.

#### 3.1.2. Sensory Evaluation of Peony Mead During Storage

The systematic sensory evaluation of flavor characteristics (visual, taste, and olfactory properties) was conducted to further investigate their development during the aging process of peony mead. The taste sensory attributes (sourness, sweetness, bitterness, and astringency) were correlated with physicochemical indicators (TSS, RS, TA, and TPC). As shown in Figure 2A, there were no significant changes in color values or wine legs in terms of visual properties. Sweetness, astringency, and bitterness gradually decreased with increasing storage time, whereas sourness initially increased but then decreased. In terms of olfactory properties, different aroma characteristics of the wine showed dynamic changes. Floral aroma, fruit aroma, sour aroma, bouquet, and lipid aroma initially weakened but then gradually increased. Fruit aroma scores significantly decreased from M1 to M2 and remained stable from M2 to M3. These results demonstrated that volatile metabolites underwent a dynamic balance regulatory mechanism during aging, shaping the complex and variable flavor profile of peony mead.

Figure 2B depicts the correlation analysis between taste sensory attributes (sourness, sweetness, bitterness, and astringency) and physicochemical indicators (TSS, RS, TA, and TPC). Sourness was significantly positively correlated with TA (*p* < 0.05) and negatively correlated with sweetness and TSS (*p* < 0.05), which indicated the dominant role of TA in sourness sensory attributes. Sweetness exhibited a significant positive correlation with TSS (*p* < 0.05) and a negative correlation with TA (*p* < 0.05), suggesting a masking effect of sweetness on sourness in the correlation between sourness/sweetness and TSS. Astringency sensation was significantly positively correlated with TPC (*p* < 0.05). High concentrations of sweet substances or elevated TSS levels can diminish the prominence of sourness in the overall flavor, revealing an important balance mechanism between sweet and sour sensations [28]. In addition, the positive correlation between sweetness and TSS underscored the contribution of sugars in soluble solids to the sweet sensation. Moreover, the negative correlation between sweetness and TA supported the critical role of the sweet–acid balance in flavor modulation. Elevated TA levels may increase sourness, thereby reducing perceived sweetness [15]. Additionally, the positive correlation between astringency and TPC highlighted the importance of the total phenolic content in astringency. Phenolic compounds bind with salivary proteins, causing dryness or astringency in the mouth, which intensifies with higher TPC levels [29].

### 3.2. Analysis of Volatile Compounds in Peony Mead During Storage

Volatile compounds in peony mead stored for different periods were analyzed via GC-IMS and principal component analysis (PCA). As shown in Figure 3A, the contribution rates of the first and second principal components are 78.4% and 16.7%, respectively, resulting in a cumulative contribution rate of 96.7%. This indicates that the first two principal components effectively capture the majority of the characteristic differences among the sample groups M1, M2, and M3. Notably, while there is partial overlap between M1, M2, and M3 in the PCA plot, M2 exhibits distinct characteristics compared to M1 and M3. This suggests that although the three groups share similarities in their principal components, M2 demonstrates a unique trend of variation. GC-IMS differential comparison chromatograms (Figure 3B,C) revealed there were significantly different volatile compounds in the storage periods. From M1 to M2, the levels of certain compounds decreased, whereas those of the other compounds increased; this trend reversed from M2 to M3. These changes were due to the dynamic equilibrium of chemical reactions in a micro-oxygen environment, including dissociation–polymerization, hydrolysis–esterification, and oxidation-reduction processes. Through aroma fingerprint analysis (Figure 3D), a total of 23 volatile metabolites were identified, of which 17 were clearly characterized. The circular heatmap further revealed that group M2 exhibited significant differences compared to M1 and M3, which is consistent with the earlier findings (Figure 4A). Based on functional group classification, these volatile metabolites were categorized as follows: alcohols (29%), esters (41%), carboxylic acids (6%), aldehydes (6%), and ketones (18%).

#### 3.2.1. Dynamic Changes in Volatile Metabolites

After 30 days of aging under micro-oxygenation, low-temperature, and low-pH conditions, the volatile metabolites in peony mead significantly changed, as demonstrated in Figure 4A. Volatile alcohols, especially 2-methyl-1-propanol and 3-methyl-1-butanol, exhibited notable dynamic changes and a reversible equilibrium between monomers, dimers, and polymers. Dissociation reactions dominated in the period of M1 to M2, leading to the decomposition of the polymers into monomers and dimers. As oxygen was depleted and the system stabilized in the period of M2 to M3, polymerization reactions increased, yielding a higher polymer content, which was regulated by temperature, pH, and oxygen concentration [18,30,31]. Notably, dimers or polymers of these higher alcohols are unique to peony mead, serving as characteristic markers.

Volatile esters, such as ethyl hexanoate and isoamyl acetate, experienced dynamic changes driven by hydrolysis–esterification equilibrium. Hydrolysis reactions predominated in the period of M1 to M2, causing a significant decrease or disappearance of these esters. As the system stabilized and the substrate supply became abundant in the period of M2 to M3, the intensification of esterification reactions increased the content of these compounds [32,33,34]. Notably, the formation rate of ethyl hexanoate was relatively slow, which was related to the low initial concentration of its substrate, hexanoic acid, and the specific storage conditions [35]. Volatile aldehydes, exemplified by 2-methyl propanal, underwent redox processes. Under microaerobic conditions, the aldehyde was oxidized to isobutyric acid, which was subsequently esterized with ethanol to form ethyl acetate. This process can be catalyzed by metal ions, such as iron ions [36,37].

Among volatile ketones, only 3-nonanone significantly decreased over the aging period because of its high volatility. Similarly, volatile acids (acetic acid) gradually declined, which was due to their participation in esterification reactions or their volatility [32,34].

#### 3.2.2. Effects of Volatile Metabolites on the Flavor of Peony Mead

The volatile flavor (olfactory attributes) of aged peony mead is derived from multiple compounds that interact to form complex and unique aroma characteristics [38]. To explore the contributions of these metabolites, a correlation analysis was conducted between the volatile metabolites identified via GC-IMS and the olfactory sensory attributes, as shown in Figure 4B.

The 3-methyl-1-butanol dimer retained fruity and banana-like flavors in monomeric form, which was significantly correlated with flowery changes [39]. Fruity changes were related to ethanol, acetone, and 6-methylhept-5-en-2-one. Ethanol contributed slightly to sweetness and alcohol flavor; acetone provided banana and apple notes; and 6-methylhept-5-en-2-one supplied floral, apple, and green bean-like flavors, increasing the complexity of the fruity attribute [40]. Similarly, bouquet changes were associated with ethanol, 6-methylhept-5-en-2-one, and 3-methyl-1-butanol dimers. Changes in acidity were correlated with acetic acid, 3-methyl-1-butanol dimers, and 3-nonanone. Acetic acid enhanced sour because of its low olfactory threshold of 0.013 mg/m^3^ [18,30,31]. Creamy changes were assigned to 3-nonanone and acetic acid, which affected creamy and acidity characteristics, even at low concentrations. Notably, some metabolites possess synergistic or antagonistic effects. The fruity flavors of 3-methyl-1-butanol dimer and acetone can cooperate to enhance floral and fruity attributes, whereas the vegetal flavor of 6-methylhept-5-en-2-one combined with the banana-like aroma of 3-methyl-1-butanol dimer increases complexity [41]. These findings highlight the intricate interactions among volatile compounds that significantly influence the sensory characteristics of aged peony mead.

### 3.3. Analysis of Nonvolatile Compounds in Peony Mead During Storage

Nonvolatile metabolites in peony mead at different storage periods were analyzed via UHPLC-MS/MS. As shown in Figure 5A, orthogonal partial least squares discriminant analysis (OPLS-DA) revealed small intragroup differences but significant intergroup differences, with clear clustering trends in the principal component space, indicating distinct metabolic profiles at different storage stages. The validation of the partial least squares discriminant analysis (PLS-DA) model in Figure 5B shows that through cross-validation, the intersection point of the Q^2^ regression line with the *y*-axis is less than 0, which confirms that the model has robust predictive performance and reliability.

The volcano plots in Figure 5C,D indicate that significant differences in metabolites (*p* < 0.05, VIP > 1, Student’s *t*-test) based on statistical significance and variable importance in projection are 35 from M1 to M2 and 23 from M2 to M3, which is ascribed to the substrate saturation effect. Higher substrate concentrations promoted more active chemical reactions in the early stage (M1 to M2), while the variation in nonvolatile metabolites markedly diminished in the later stages (M2 to M3) as the substrates were progressively depleted and the system stabilized [34,42]. Based on their structural and functional properties, these metabolites are classified into six categories: sugars and their derivatives, organic acids and their derivatives, flavonoids and their derivatives, amino acids and their derivatives, lipids and their derivatives, and other compounds.

#### 3.3.1. Organic Acids and Their Derivatives

Figure 6A illustrates the substantial transformations in organic acids and their derivatives during the M1-M2 period. D-galactarate levels decreased, possibly due to its involvement in redox or decarboxylation reactions [43]. The contents of glycolate and 2-dehydro-3-deoxy-D-gluconate increased, reflecting the accumulation of carbohydrate degradation intermediates [44]. 3-Hydroxyglutaric acid levels increased under the drive of fatty acid oxidation [45]. The homocitrate level decreased, which was associated with condensation or dehydration reactions [46]. The loganic acid levels increased, indicating the accumulation of its precursor molecules through nonenzymatic conversions [47].

The changes in compounds exhibited a distinct trend in the later stage (M2–M3). The glucosinolate concentration decreased, possibly due to its conversion into volatile acids or the formation of complex aroma molecules [48]. (*S*)-2-hydroxyglutarate levels increased, which was related to the redox balance between lactic acid and α-ketoglutaric acid, increasing the smoothness of the wine [49]. Aromatic metabolites such as 3-1-NapCO-Ind-PenA accumulated, contributing to aroma complexity through the degradation of aromatic amino acids [50]. The abundance of metabolites, such as larixinic acid decreased, which was potentially ascribed to thiol-related condensation reactions [51]. Therefore, organic acids and their derivatives improved the rich and harmonious flavor profiles of the peony mead through reactions such as condensation, dehydration, and redox reactions.

#### 3.3.2. Carbohydrates and Their Derivatives

Figure 6B demonstrates significant decreases in oligosaccharides (e.g., melezitose, maltotetraose) and glycosides (e.g., plantamajoside) under acid hydrolysis and oxidative degradation during M1–M2. This process generated small organic acids, aldehydes, ketones, and free non-sugar groups, which collectively enhanced wine complexity [52]. The increase in the nodakenin levels suggested that some glycosides may be formed through nonenzymatic pathways, which facilitated the development of flavor. The content of polyols (e.g., D-arabitol) significantly increased during the later stage (M2–M3), which was associated with the degradation of oligosaccharides and the reduction in polyhydroxy aldehyde, improving the roundness of the taste [53]. The accumulation of bound polyphenols (e.g., resveratrol 4′-*O*-D-glucuronide) enhanced molecular stability and optimized flavor balance. The decrease in the level of petunidin 3-galactoside indicated the degradation of anthocyanin, which may introduce new aroma layers. The aging process of peony mead was characterized by the consumption of oligosaccharides and glycosides in the initial stage (M1–M2), followed by the accumulation of polyols and bound polyphenols in the later stage (M2–M3).

#### 3.3.3. Flavonoids and Their Derivatives

Figure 6C highlights a notable rise in tamarixetin and related flavonoids during M1-M2. This increase was driven by oxidation-induced methylation or hydroxylation processes that enhanced molecular stability while enriching flavor characteristics [54]. Furthermore, an increase in the concentration of khelloside indicated nonenzymatic glycosylation, which contributed to a sweet and smooth taste. In the later stage (M2–M3), the contents of flavonoids such as rutin, 2′,3′-dimethoxyflavone, and homoplantaginin decreased, which was caused by hydrolysis, deglycosylation, or oxidative degradation. Rutin degraded into quercetin and participated in polymerization reactions [55,56]. Elevated levels of Macluraxanthone suggested that the cross-linking or condensation of flavonoid molecules formed high-molecular-weight products, increasing their antioxidant capacity, color stability, and flavor complexity [57]. The dynamic changes in flavonoids and their derivatives during aging were regulated by chemical reactions, such as methylation, hydrolysis, deglycosylation, and cross-linking, which affected the antioxidant capacity, color stability, and flavor complexity.

#### 3.3.4. Amino Acids and Their Derivatives

As illustrated in Figure 6D, L-carnitine levels declined noticeably during M1–M2 due to oxidation or carbonylation reactions. These processes generated secondary metabolites that enriched the wine’s flavor profile [58]. Maillard reactions or polyphenol cross-linking reduced the Gly-His-Lys levels, resulting in the formation of pyrazines and heterocyclic compounds. DL-glutamic acid levels decreased through decarboxylation into γ-aminobutyric acid, increasing the degree of smoothness of the wine [59]. The increase in pyridoxal levels indicated its stability and ability to act as a cofactor to promote the continuation of Maillard reactions, thereby increasing aroma complexity [60]. In the later stage (M2–M3), γ-L-glutamyl-L-glutamic acid significantly increased, suggesting nonenzymatic condensation of glutamic acid into dipeptides. These dipeptides enhanced umami taste and possessed antioxidant properties, improving the stability of wine [61]. In addition, the adenosine was likely consumed in hydrolysis or condensation reactions, providing precursors for volatile flavor compounds [62]. The dynamic changes in amino acids and their derivatives during aging drove the formation of rich and complex flavors and sensory characteristics via pathways such as Maillard reactions, decarboxylation, and condensation while increasing the stability of wine quality.

#### 3.3.5. Lipids and Their Derivatives

Figure 6E shows a decline in key lipid compounds, such as oleamide and sphingolipids, during M1–M2. This reduction was attributed to oxidation, hydrolysis, or cross-linking reactions. Oleamide and sphingolipids can be oxidized into short-chain aldehydes or acids, imparting specific aromas. The hydrolysis of monoglycerides released free fatty acids that were further oxidized into volatile compounds, resulting in herbal, nutty, or fruity flavors [63]. In the later stage (M2–M3), the FA 18:1+3O and octadecanoic acid levels decreased, corresponding to the oxidation and polymerization of unsaturated fatty acids, which generated volatile aldehydes, ketones, and aroma-active molecules. Stearic acid may participate in condensation reactions to form high-molecular-weight precipitates, which enhance wine clarity but reduce some flavor precursors [64,65]. The dynamic changes in lipids and their derivatives regulated the generation of secondary metabolites through oxidation, hydrolysis, and polymerization, thereby increasing the aroma complexity, texture balance, and quality stability of wine.

#### 3.3.6. Other Categories

As depicted in Figure 6F, linalool oxide levels rose during M1-M2 due to oxidation processes that enhanced floral and fruity aromas. Cuminaldehyde was involved in condensation reactions to form Schiff base intermediates, resulting in a low level. Trigonelline levels were reduced through demethylation to nicotinic acid, supporting the accumulation of flavor precursors [66,67,68]. The levels of 2-methylamino-1-phenylbutane decreased, whereas the phenylethylamine levels increased, indicating active demethylation or degradation. Phenylethylamine may participate in carbonyl–amino condensation to form volatile flavor compounds [69]. In the later aging stage (M2–M3), the 4-hydroxybenzaldehyde levels significantly increased because of lignin degradation, which contributed to a warm woody aroma. Elevated levels of niacinamide indicated the accumulation of nicotinic acid derivatives, increasing antioxidant capacity [70,71]. *p*-Methoxycinnamic acid ethyl ester levels increased, demonstrating that aromatic carboxylic acids generated pleasant floral and fruity aroma molecules through esterification [72]. These dynamic changes generated key molecules such as aromatic aldehydes, terpene derivatives, and aromatic carboxylic acids via oxidation, hydrolysis, and condensation reactions, thereby significantly enhancing the flavor complexity and sensory harmony of wine.

During peony mead aging, nonvolatile metabolites (sugars, organic acids, flavonoids, amino acids, lipids, and other compounds) underwent significant changes driven by nonenzymatic reactions such as oxidation reduction, hydrolysis, and condensation. These changes impacted the flavor characteristics, sensory quality, and functional properties of wine throughout the early (M1–M2) and later (M2–M3) stages of aging.

#### 3.3.7. Effects of Nonvolatile Metabolites on the Flavor of Peony Mead

The correlation analysis was performed on significantly differential nonvolatile metabolites to further explore the correlations between nonvolatile metabolites, taste attributes, and wine body physicochemical indicators, as depicted in Figure 7.

##### Sourness and Related Metabolites

Sourness was significantly positively correlated with 2-dehydro-3-deoxy-D-gluconate, glycolate, and xanthine but negatively correlated with homocitrate and D-pinitol. As organic acids, 2-dehydro-3-deoxy-D-gluconate and glycolate provided fruity acidity, whereas xanthine enhanced sour perception through mild bitterness and astringency [71]. In contrast, homocitrate reduced overall acidity by neutralizing free hydrogen ions, resulting in a softer taste. D-pinitol formed complexes with organic acids, weakening sour and contributing to a balanced flavor [73].

TA levels were positively correlated with glycolate and xanthine levels but negatively correlated with homocitrate and D-pinitol levels. Glycolate and xanthine promoted the formation of acidic substances through the transformation of hydroxy acid or purine derivatives, whereas homocitrate reduced acid accumulation via chemical neutralization. D-pinitol indirectly lowered titratable acid levels through oxidation or transformation processes [74].

##### Sweetness and Related Metabolites

Sweetness, similar to changes in the TSS, was positively correlated with D-Galactarate, Gly-His-Lys, petunidin 3-galactoside, and maltetraose but negatively correlated with the contents of linalool oxide and loganic acid. Lactarate enhanced the balance of sourness and sweetness at low concentrations. Gly-His-Lys stabilized sweetness molecules and enhanced sensory performance, acting as a sweetener or flavor enhancer [75]. Petunidin 3-galactoside delayed the degradation of sweetness molecules through its antioxidant activity, improving the taste balance. Maltotetraose, generated from the degradation of starch or polysaccharide, gradually increased sweetness [76,77]. Conversely, high concentrations of linalool oxide masked sweetness signals with floral aromas and affected sweet taste perception through nonenzymatic oxidation. The bitterness and strongly acidic characteristics of loganic acid weakened the taste balance and inhibited both sweetness and TSS [78].

RS levels were positively correlated with melezitose, Haploside D, and Cuminaldehyde but negatively correlated with khelloside and 3′,5′-dimethoxy-3,5,7,4′-tetrahydroxyflavone. Oligosaccharides, such as melezitose, released reducing sugars through hydrolysis, whereas aromatic compounds, such as Cuminaldehyde, participated in the Maillard reaction, generating flavor precursors [79]. Polyphenols, such as khelloside, inhibited initial Maillard reaction free radical chain reactions, reducing the consumption of reducing sugar. Additionally, polyphenols competed for metal ions or catalytic sites, lowering the efficiency of the Maillard reaction [80].

##### Bitterness, Astringency, and Related Metabolites

Bitterness had a significantly positive correlation with *S*-methyl-5′-thioadenosine (MTA) and octadecanoic acid but a negative correlation with D-arabitol. MTA, as a nucleoside compound, amplified the bitter taste signal by directly activating bitter taste receptors (such as TAS2R family members) or generating stronger bitter-tasting small molecules [81]. The hydrophobic nature of octadecanoic acid may alter the solution microenvironment or stabilize other hydrophobic bitterness molecules, thereby increasing the overall bitterness [82]. D-arabitol, as a sugar alcohol, activated sweet taste receptors and antagonized bitterness signals through its sweet characteristics, and its high polarity reduced the efficiency of hydrophobic bitter molecules [83].

Astringency was primarily derived from polyphenolic components in mead, with the total phenolic content significantly positively correlated with astringency, particularly rutin. The phenolic hydroxyl groups in rutin bound to salivary proteins via hydrogen bonds and hydrophobic interactions, yielding a dry and tight oral sensation. A low-pH environment further enhanced this binding efficiency, intensifying the puckery sensation. Rutin protects other polyphenolic components from degradation through its antioxidant activity, supporting the flavor complexity of mead [84].

TPC was significantly positively correlated with larixinic acid and rutin, which maintained high TPC levels through synergistic antioxidant effects and polymerization. Larixinic acid scavenged singlet oxygen, while rutin efficiently captured hydroxyl radicals, complementing each other to weaken polyphenol degradation [85]. Additionally, larixinic acid provided carboxyl functional groups for hydrogen bond formation, whereas rutin promoted π–π interactions through its aromatic ring structure, forming macromolecular complexes [86]. These effects increased the TPC in the short term but may lead to sediment formation and decreased apparent values during long-term aging. Moreover, the hydrolysis of rutin generated quercetin and other potent antioxidants, further protecting tannins and other flavonoids from degradation [87].

During the aging process of peony mead, organic acids regulated the balance of sourness and sweetness, polyphenols imparted astringency and stabilized the antioxidant system, lipids and nucleosides increased the complexity of the body and bitterness, and reduced sugar levels were influenced by dynamic metabolite changes. These nonvolatile metabolites collectively shaped the flavor complexity, taste balance, and physicochemical stability of honey mead.

## 4. Discussion

The physicochemical properties, sensory properties, and volatile and nonvolatile metabolites of peony mead were systematically analyzed during short-term aging, revealing the dynamic changes in its flavor characteristics and underlying regulatory mechanisms.

First, key physicochemical indicators (TSS, RS, TA, and TPC) had significant changes during the short-term aging process. Correlation analysis between these indicators and sensory attributes (sourness, sweetness, astringency, and bitterness) highlighted the critical role of the sweet–acid balance and polyphenol content in flavor perception. The negative correlation between sweetness and acidity confirmed their competitive effect at the sensory level. These findings suggested that adjusting the initial sugar concentration or acidity during the production process effectively optimized the flavor balance. However, the current study was limited to 30 days of short-term storage, which made it difficult to fully assess the impact of long-term aging on these indicators or their correlations. Future research should extend the storage time and incorporate more complex environmental variables (e.g., temperature and oxygen concentration) to comprehensively evaluate the dynamic balance between physicochemical parameters and sensory attributes during aging.

Second, the analysis of volatile metabolites revealed significant and complex dynamic changes in key aroma molecules such as alcohols, esters, and aldehydes. Alcohols exhibited polymerization–dissociation equilibrium, whereas esters were regulated by hydrolysis–esterification reactions, collectively shaping the aroma profile of the wine. Owing to the numerous types and low concentrations of volatile metabolites, their olfactory contributions were influenced by complex synergistic effects. The contributions of key metabolites (e.g., 3-methyl-1-butanol dimer and 6-methylhept-5-en-2-one) to olfactory (e.g., floral aroma and fruit aroma) were initially identified using GC-IMS technology combined with PCA and correlation analysis. However, further exploration is needed to understand potential synergistic or antagonistic interactions among different metabolites. Sensory verification should be conducted on single compounds and mixtures to clarify their actual olfactory contributions and explore the molecular mechanisms of degradation or transformation pathways under various storage conditions.

Regarding nonvolatile metabolites, UHPLC-MS/MS analysis identified six major categories of significantly different metabolites, including sugars, organic acids, flavonoids, and others. These findings highlight the distinct metabolic profiles among the samples. These metabolites promoted the development of flavor characteristics through nonenzymatic reactions such as oxidation, hydrolysis, and condensation. However, owing to the complexity and diversity of nonvolatile metabolites, the molecular mechanisms behind these dynamic changes remained incompletely understood. Future studies should track the transformation pathways of key substrates and intermediate products and quantify change patterns by combining reaction kinetics models.

In conclusion, the dynamic balance of volatile compounds (e.g., alcohols and esters) and the contribution of nonvolatile metabolites (e.g., sugars, organic acids, and flavonoids) to flavor characteristics were analyzed via GC-IMS and UHPLC-MS/MS technologies. These results lay a solid scientific foundation for refining production processes while enhancing honey mead quality. Future research should focus on long-term storage and molecular mechanisms for more in-depth exploration.

## Figures and Tables

**Figure 1 foods-14-01021-f001:**
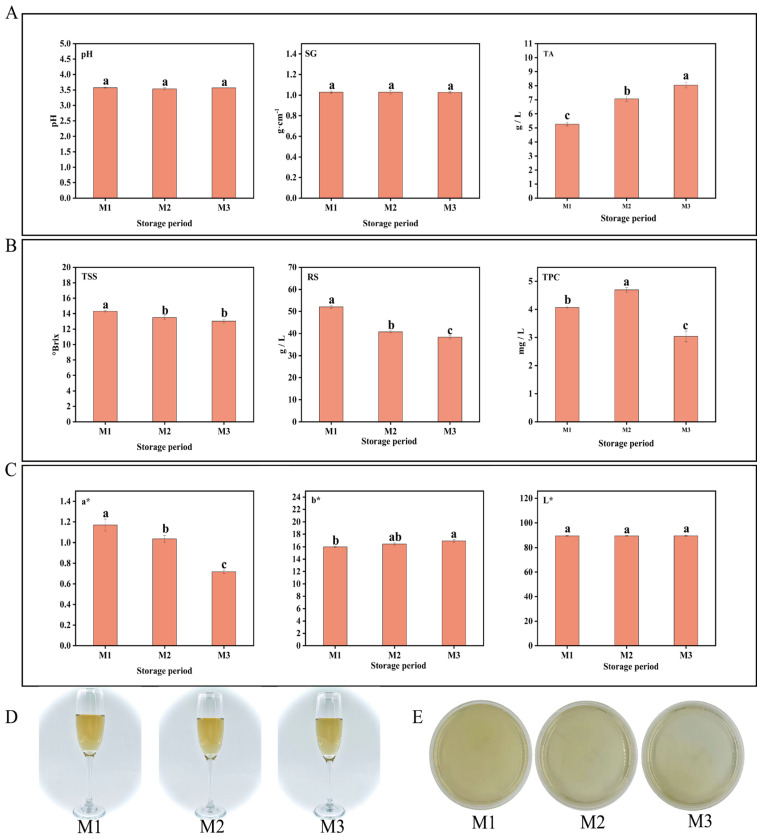
pH, SG, and TA (**A**). TSS, RS, and TPC (**B**). Color parameters (**C**). Wine body image (**D**). Colony count (**E**). Different lowercase letters above bars indicate significant differences (*p* < 0.05, one-way ANOVA with Tukey’s HSD test) (**A**–**C**).

**Figure 2 foods-14-01021-f002:**
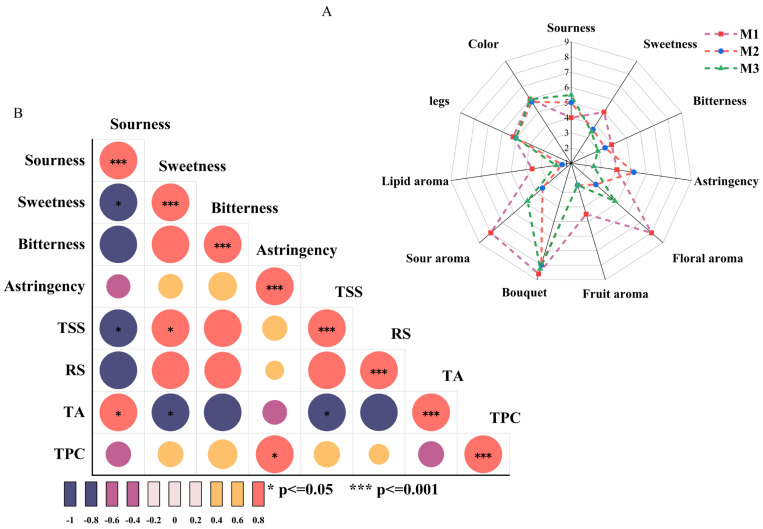
Sensory assessment radar map (**A**) and thermal map of the correlations between sensory attributes and physical/chemical indices (**B**).

**Figure 3 foods-14-01021-f003:**
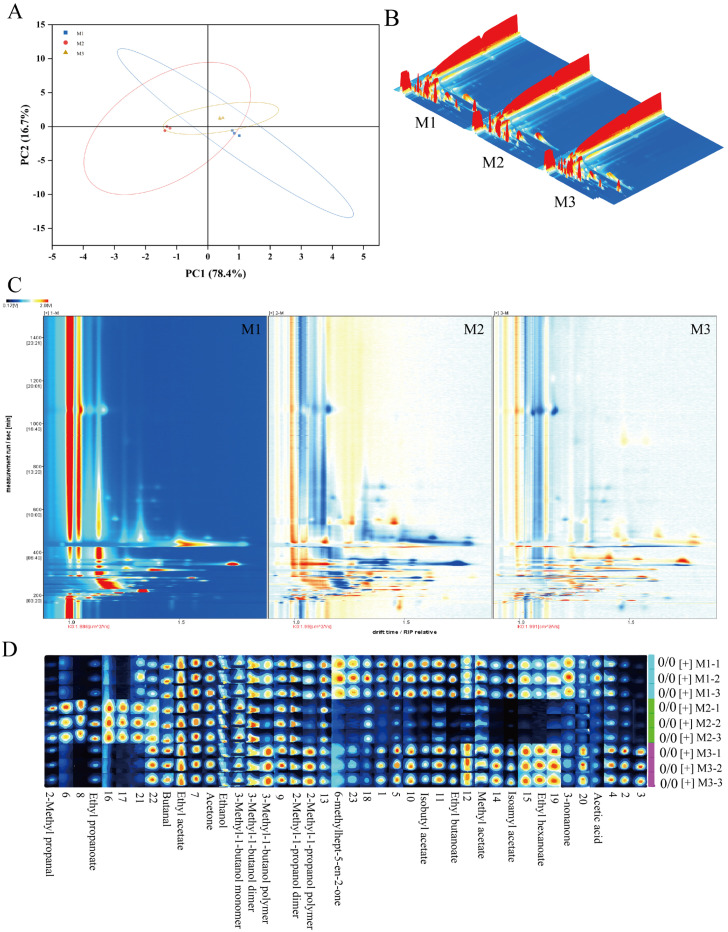
Principal component analysis diagram (**A**), GC-IMS difference comparison bird’s-eye view, Color intensity: white for low, red for high, darker for greater concentration (**B**), GC-IMS difference comparison chromatogram, Color intensity: white for low, red for high, darker for greater concentration. (**C**), and aroma fingerprint diagram of peony mead, Color intensity: white for low, red for high, darker for greater concentration. (**D**).

**Figure 4 foods-14-01021-f004:**
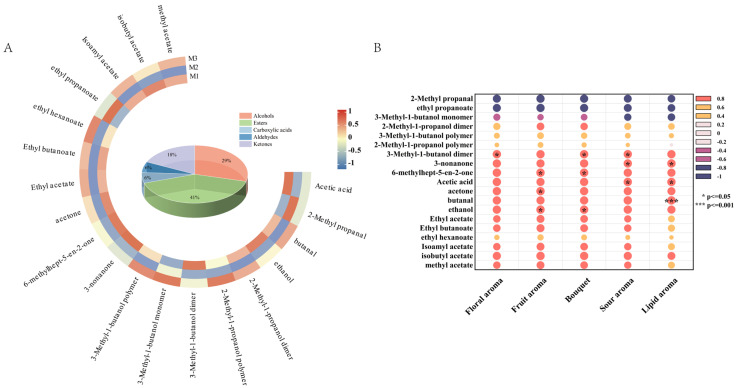
Heatmap of identifiable volatile metabolites (**A**) and correlation analysis diagram between olfactory attributes and volatile metabolites (**B**).

**Figure 5 foods-14-01021-f005:**
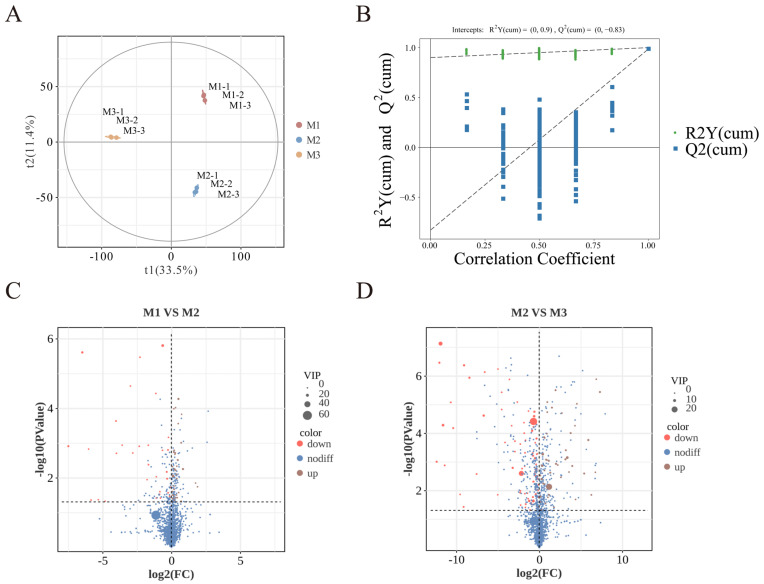
PLS-DA diagram (**A**). Permutation test diagram (**B**). Volcano plot of M1 vs. M2 (**C**). Volcano plot of M2 vs. M3 (**D**).

**Figure 6 foods-14-01021-f006:**
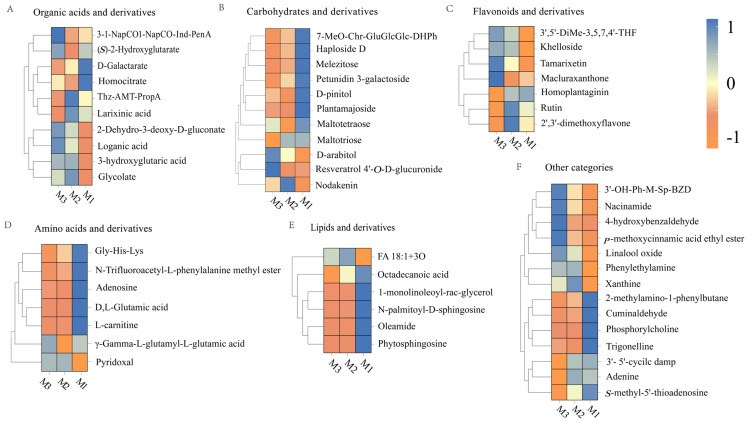
Heatmap of organic acids and derivatives (**A**), carbohydrates and derivatives (**B**), flavonoids and their derivatives (**C**), amino acids and derivatives (**D**), lipids and their derivatives (**E**), and other categories (**F**).

**Figure 7 foods-14-01021-f007:**
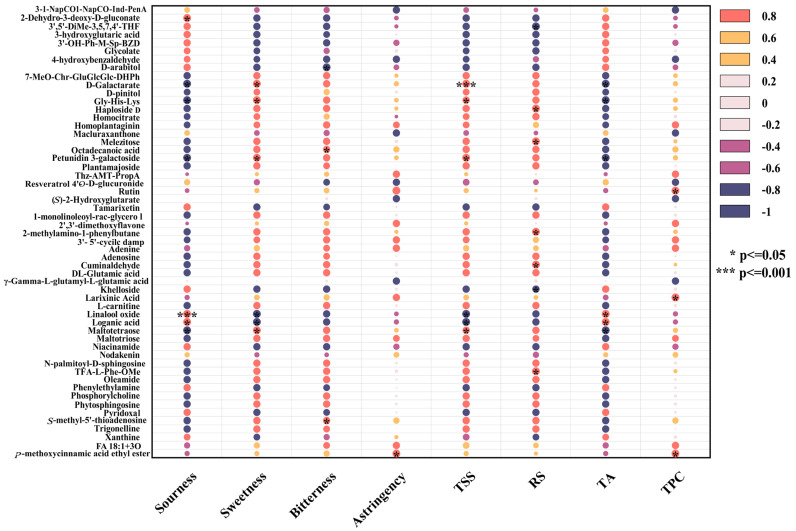
Correlation analysis diagram between sensory properties and nonvolatile metabolites.

**Table 1 foods-14-01021-t001:** Color difference (Δ*E*) among peony mead in different storage periods (M1–M3).

	M1	M2	M3
M1	0.00		
M2	0.479	0.00	
M3	1.017	0.551	0.00

## Data Availability

The original contributions presented in this study are included in the article. Further inquiries can be directed to the corresponding author.

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
