# Peer review of "Metabolic Dynamics and Sensory Impacts of Aging on Peony Mead: Insights into Nonenzymatic Reactions"

_foods, 2025, doi:10.3390/foods14061021_

Round 1

Reviewer 1 Report

Comments and Suggestions for Authors

In this research the authors investigated sensory properties of volatile and nonvolatile metabolites of peony mead. Analytical approaches were mostly applied in terms of chromatography and mass spectrometry (GC-IMS and UHPLC-MS/MS). A sensory analysis was also performed. The manuscript is in general well written, however it needs a complete reveal and re-work to avoid the following flaws:

Line 42: Please check spaces.

Line 44: The information about “Western Zhou Dynasty” is not so clear for many readers.

Line 45. Same for “Compendium of Materia Medica written by 44 Li Shizhen in the Ming Dynasty“

Line 57-59 – Please review the sentence “Short- term aging not only improves flavor and balance taste in a short time but also accelerates the attainment of a drinkable state for peony mead, allowing it to be brought to the market“

Line 60. Why beer is here mentioned “give the beer a”?

Line 83. Please check here and throughout the manuscript that species names like “Saccharomyces cerevisiae” is in italic.

Line 96… Please separate city and country with comas like in “Chengdu China”

Line 161. Normally gradients are described with an initial and final concentration. “The gradient was 95% B for 0.5 min“ this should be an isocratic flow.

Please improve the quality of figure 1. It is almost not possible to read also the embedded text.

Figure 3. please improve the quality of figure and the size of the text.

The same for Figure 5. It is not possible to read the text with clarity.

Reviewer 2 Report

Comments and Suggestions for Authors

The manuscript foods-3501985, “Metabolic Dynamics and Sensory Impacts of Aging on Peony mead: Insights into Nonenzymatic Reactions”, provided interesting observations about quality of Peony mead. However, the presentation quality of this manuscript was low. So, I think that this review should be rejected. My comments are as follows:

Line 2, 14, and 45:

The ‘Peony’ should be indicated together with scientific name, .

Line 17:

The ‘GC-IMS’ and ‘UHPLC-MS/MS’ should be indicated together with both formal names.

Line 83 and 102-103:

The scientific name of yeast ‘Saccharomyces cerevisiae’ must be indicated in Italic.

Line 92-98:

These measurement devises were the ‘Instruments’, not the ‘Materials’.

Line 146-147:

What is the unit of measured VOCs? mg/L? or peak area? If the authors used this data for statistical analysis, semi-quantification method via GC-IMS must be explained more detail. If not, readers couldn’t evaluate the validity of this study.

2.7 UHPLC‒MS/MS analysis:

The semi-quantification method of metabolites via UHPLC-MS/MS wasn’t explained.

All figures:

The graphic resolution of all figures was very poor. Readers couldn’t recognize any data necessary for understanding this study.

Comments on the Quality of English Language

No additive comment.

Round 2

Reviewer 2 Report

Comments and Suggestions for Authors

The manuscript foods-3501985 v2, “Metabolic Dynamics and Sensory Impacts of Aging on Peony mead: Insights into Nonenzymatic Reactions”, provided interesting observations about quality of Peony mead. I checked all revised points. I think that the presentation quality of this manuscript became drastically improved, and that this manuscript could be accepted after minor revision. My comments are as follows:

<compounds names>

-Line 398 and Figure 6 and 7

(S)-2-hydroxyglutarate

-Line 417 and Figure 6 and 7

4'-O-D-glucuronide

-Line 534 and Figure 6 and 7

S-methyl-5'-thioadenosine (MTA)

In all text and figures, the ‘S’ and ‘O’ must be indicated in Italic.

-Line 461 and Figure 6 and 7

What is ‘Fa 18:1+3o’ means? This abbreviation didn’t define anywhere in this manuscript. If this means the mass spectrum ‘FA 18:1+3O’, please explain this or another name should be defined, for example ‘oxidized fatty acid ‘FA 18:1+3O‘’, etc. In addition, the ’FA‘ and ‘O’ must be indicated in the capital letter, because these means ’Fatty acid‘ and ‘Oxygen’, respectively.

-Line 480 and Figure 6 and 7

The name ‘P-Methoxycinnamic acid ethyl ester (P-methoxycinnamic acid ethyl ester)’ must be indicated as ’p-Methoxycinnamic acid ethyl ester (p-methoxycinnamic acid ethyl ester)’. In addition, the ‘p’ must be indicated in Italic.

- Figure 6 and 7

Resveratorol 4’-o-D-gulcuronide

The ‘o’ must be indicated the capital letter ‘O’ in Italic.

- Figure 6 and 7

Dose the term ‘&apos’ mean apostrophe? If so, these are should be indicated as the symbol.

Comments on the Quality of English Language

The manuscript contains careless errors in English grammar.
